# Augmenting Nutrient Acquisition Ranges of Greenhouse Grown CBD (Cannabidiol) Hemp (*Cannabis sativa*) Cultivars

**Jennifer Kalinowski** [1,2,*] **, Keith Edmisten** [2] **, Jeanine Davis** [1] **, Michelle McGinnis** [3] **, Kristin Hicks** [3] **, Paul Cockson** [1] **, Patrick Veazie** [1] **and Brian E. Whipker** [1]

[1] Department of Horticultural Science, North Carolina State University, Raleigh, NC 27695, USA; jeanine_davis@ncsu.edu (J.D.); pncockso@ncsu.edu (P.C.); phveazie@ncsu.edu (P.V.); bwhipker@ncsu.edu (B.E.W.)

[2] Department of Crop and Soil Science, North Carolina State University, Raleigh, NC 27695, USA; kledmist@ncsu.edu

[3] North Carolina Department of Agriculture and Consumer Services, Raleigh, NC 27601, USA; michelle.mcginnis@ncagr.gov (M.M.); kristin.hicks@ncagr.gov (K.H.)

* Correspondence: jmkalino@ncsu.edu

**Abstract:** There is a growing interest in the production of hemp for the extraction of cannabidiol (CBD) due to reported therapeutic benefits. Recent policy reform has permitted state hemp pilot programs, including the land grant research institutions, the ability to investigate the potential of growing and harvesting *Cannabis sativa* plants (≤0.3% tetrahydrocannabinol) for these purposes in the U.S. There are vast gaps of knowledge regarding the fertility requirements of hemp cultivars grown in a horticultural production setting for floral attributes such as the cannabinoid constituents. Foliar tissue analysis provides an avenue to determine adequate ranges for nutrient uptake and estimating fertilizer requirements prior to visual symptoms of deficiency or toxicity. To facilitate a survey range of elemental nutrient acquisition in hemp cultivars propagated for CBD production, foliar analysis was executed using the most recently mature leaves (MRML) of mother stock plants. All plants were maintained in the vegetative stage for twelve weeks, prior to initiation of cutting for clone harvesting. A total of thirteen cultivars were utilized to broaden previously reported baseline survey ranges. Significant differences were found among all thirteen cultivars in accumulation of both micro and macro essential nutrients, widening the range of the fertility requirements of *Cannabis* plants grown in this production model for CBD harvesting.

**Keywords:** foliar; macronutrients; micronutrients; deficiency; foliar analysis; fertility; toxicity; indica; subspecies

## 1. Introduction

*Cannabis sativa* is a multipurpose crop with a long history of use, dating back millennia with cultivation for fiber, seed, oil, and its medicinal/psychoactive cannabinoid constituents [1,2]. Cannabinoids are secondary metabolites obtained primarily from the inflorescence or infructescence of the *Cannabis* plant. There are over 100 cannabinoids found to be present in the floral structures of the plant [3,4]. The most abundant cannabinoids identified are tetrahydrocannabinol (THC) and cannabidiol (CBD). THC is a cannabinoid well known for its psychoactive effects and CBD is a non-intoxicant with reported therapeutic benefits [3,5,6].

Recent changes in legislation both federally and at the state level have redefined the legality of *Cannabis* and the distinction between marijuana, a Scheduled I drug, and hemp. The definition of



hemp according to the Hemp Farming Act 2018 [7] is *Cannabis sativa* containing 0.3% or less of THC in any part of the plant with no regulation on the amount of CBD.

These policy reforms paired with research on the therapeutic benefits of CBD have resulted in heightened interest among U.S. farmers and consumers, thereby opening the market for the growing and harvesting of hemp for the non-intoxicant cannabinoid CBD [5]. A recent survey conducted among North Carolina farmers [8] showed that 52% are interested in growing hemp for CBD production. However, commercial hemp plant breeding programs have been limited and have mainly focused on field fiber or grain production over the past 50 years [9]. Cultivation of hemp for floral production has evolved to follow more of a horticultural framework in a greenhouse or bedded field production system. Many farmers are unfamiliar with the production practices for cultivating the crop in these systems [9]. One aspect of uncertainty likely stems from the lack of reliable information on fertility requirements of the hemp plant in horticultural production models in addition to nutrient acquisition ranges to promote optimal growth and yield while aiding fertility management strategies.

Baseline survey ranges for *Cannabis* developed by Bryson et al. [10] for greenhouse nursery production and most recently by Landis et al. [11] for greenhouse mother stock used for propagation of hemp cultivars for CBD production, have provided a starting point for measures of hemp nutrient status. Foliar leaf tissue values obtained from these studies lend critical information in both fertility management and diagnosis of nutrient disorders as illustrated by Cockson et al. [12], wherein evaluations of deficiency and toxicity ranges of *Cannabis sativa* "T1" were established by leaf tissue concentration thresholds.

The aim of this study was to expand on fertility survey ranges for greenhouse-grown hemp stock plants propagated for CBD production by including a larger sample size and a wider range of cultivars. This information will aid researchers, agricultural extension agents, and producers in gauging appropriate fertility requirements of hemp strains, thus enabling early intervention prior to the appearance of visual nutrient disorder symptoms when utilizing leaf tissue analysis.

## 2. Materials and Methods

Thirteen hemp cultivars ("BaOx", "Cherry 2.0", "Cherry Citrus", "Cherry Cross", "Cherry Wine", "Cherry Cross × Cherry Wine", "Early Pearly", "Electra", "Endurance", "Midwest", "Stout", "Suver Haze", and "Sweetened") were grown as stock plants in a greenhouse for CBD-hemp transplant production. Cultivars were chosen based on commonality of use in greenhouse and indoor production in our region. Twelve plants per cultivar were grown in 11.4 L pots containing a peat-based substrate (Sunshine Mix #1, Sun Gro Horticulture, Agawam, MA, USA) irrigated with a complete fertilizer (13–2–13 Calcium-Magnesium (Ultrasol TM SQM, North America Corporation, Atlanta, GA, USA)) supplied at 150 mg $L^{-1}$ N. Fertilizer was applied through an automatic irrigation system that ran daily for three minutes. Foliar tissue samples of the most recently mature leaves (MRML) were collected from 12-week-old mother stock plants prior to harvesting of cuttings for clonal propagation. Plants were maintained in the vegetative state of development with night interruption lighting from 22:00 to 2:00 h. Plants were healthy and did no exhibit any signs of nutrient distress.

A total of twenty MRML (first fully expanded leaves, usually at the third or fourth internode from the top of the plant) were collected from the shoots of three plants per cultivar to produce a single replicate, with four replicates harvested per cultivar, for a total of 52 leaf tissue samples. The leaf tissue samples were rinsed with distilled (DI) water, washed in a solution of 0.5 M HCl, and rinsed again in DI water. The leaf tissue samples were then dried at 70 °C for 48 h. Dry weights were measured, and the samples were ground in a mill (Thomas Wiley® Mini-Mill, Thomas Scientific, Swedesboro, NJ, USA) with a 20-mesh (1 mm) screen. Ground tissue samples were analyzed for nutrient concentrations by the North Carolina Department of Agriculture and Consumer Services (NCDA&CS) Agronomic Division [13]. Total N concentration was determined by oxygen combustion gas chromatography with an elemental analyzer (NA1500s2; CE Elantech Instruments; Lakewood, NJ, USA) (AOAC 1990; Campbell 1992) on a 0.5 g aliquot of the dried and ground sample. Results are expressed in percent (%)

on a dry-weight basis [13]. Total concentrations of P, K, Ca, Mg, S, Fe, Mn, Zn, Cu, B, and Al were determined by Inductively Coupled Plasma-Optical Emission Spectrometry (ICP-OES) (Spectro Arcos EOP and Arcos II EOP, Spectro Analytical: A Division of Ametek; Mahwah NJ), after closed-vessel nitric acid ($HNO_3$) digestion in a microwave digestion system (MARS 6 Microwaves; CEM Corp.; Matthews, NC) [13]. Total N, P, K, Ca, Mg, and S are expressed as a percentage (%) and Fe, Mn, Zn, Cu, B, and Al are expressed in (mg/kg) on a dry weight basis (Supplementary Materials S1). Data were analyzed with SAS version 9.4 (SAS Institute, Cary, NC, USA) and subjected to analysis of variance (ANOVA) using PROC GLM. Significant differences among cultivars were evidenced by F-tests ($p \leq 0.05$) and Tukey's HSD test was used to determine significant differences among cultivar means ($p \leq 0.05$).

## 3. Results

### 3.1. Macronutrient Leaf Concentration

There were significant differences in primary and secondary macronutrient concentration means among all 13 cultivars (Table 1). The minimum nitrogen (N) concentrations for all cultivars were above the lowest reported value by Landis et al. [11] of 2.65% to 4.47% (Table 2). One cultivar, "Midwest", had a mean of 4.82% that exceeded the maximum concentration level of 4.76% reported by Bryson et al. [10] (Table 2) Nutrient concentrations for phosphorus (P) fell within the minimum reported concentration of Bryson et al. [10] and maximum concentration levels were reported by Landis et al. [11] across all cultivars. Potassium (K) nutrient concentrations were above the 1.54% minimum value previously reported [11] but three cultivars ("Cherry 2.0", "Endurance", and "Midwest") exceeded the maximum reported concentration of 2.98% [11], at concentrations of 3.18%, 3.41%, and 3.06% respectively. Additionally, the average K mean across all cultivars was found to be 2.57%, higher than the 2.42% mean reported by Landis et al. [11] Calcium (Ca) concentrations were within the minimum levels reported by Landis et al. [11], but four cultivars ("Cherry Citrus", 'Cherry Wine", "Early Pearly", and "Endurance") exceeded the maximum level of 4.42% reported by Bryson et al. [10] with maximum concentrations of 5.16%, 4.95%, 5.34%, and 4.93%, respectively. Magnesium (Mg) levels were found to be within the minimum concentrations reported by Landis et al. [11] and below the maximum concentration reported by Bryson et al. [10] Though sulfur (S) concentrations were found to be above the lowest minimum value of 0.17% reported by Bryson et al. [10], concentrations exceeded the maximum value of 0.29% reported by Landis et al. [11] for twelve of thirteen cultivars, with an average mean of 0.31% across all cultivars.

Macronutrient mean comparisons using Tukey's HSD test of the thirteen cultivars identified significant differences among cultivar means for all macronutrient concentrations (Figure 1). The mean N concentration was found to be significantly higher in "Early Pearly", "Electra", and "Midwest" compared to "Cherry Cross × Cherry Wine", "Suver Haze", and "Sweetened" with an MSD (Minimum Significant Difference) $\geq 0.58\%$. "Midwest" and "Stout" were found to be significantly greater in P compared to "Cherry Cross × Cherry Wine" with an MSD $\geq 0.08\%$. "Cherry 2.0" and "Endurance" K means were similar to "Early Pearly", "Midwest", and "Suver Haze" but higher than other cultivars with an MSD $\geq 0.39\%$. "Early Pearly" had similar Ca concentration means as "Cherry Citrus", "Cherry Wine", and "Endurance" but was higher than all other cultivars with an MSD $\geq 1.29\%$. "Cherry Wine" had a higher Mg concentration than "BaOx", "Early Pearly", "Electra", "Stout", and "Sweetened" with an MSD $\geq 0.14\%$. Sulfur concentration was highest in "Cherry Citrus" and most statistically different from "Suver Haze" and "Sweetened" with an MSD $\geq 0.05\%$.

**Table 1.** Leaf tissue macronutrient concentration ranges for thirteen greenhouse-propagated cannabidiol (CBD) hemp cultivars. Means are delineated in parentheses following each cultivar nutrient range. Plants were 12 weeks old and in a vegetative stage of development prior to the excision of initial cuttings for propagation.

| Cultivar | Macronutrient (%) | | | | | |
|---|---|---|---|---|---|---|
| | N | P | K | Ca | Mg | S |
| BaOx | 4.25–4.52 (4.36) | 0.33–0.38 (0.36) | 2.23–2.43 (2.37) | 1.66–2.37 (2.09) | 0.34–0.44 (0.41) | 0.28–0.32 (0.30) |
| Cherry 2.0 | 4.03–4.35 (4.22) | 0.32–0.35 (0.34) | 2.83–3.18 (2.98) | 2.38–3.14 (2.60) | 0.48–0.55 (0.51) | 0.28–0.30 (0.29) |
| Cherry Citrus | 3.87–4.24 (4.09) | 0.31–0.37 (0.34) | 2.30–2.75 (2.50) | 3.01–5.16 (4.04) | 0.38–0.61 (0.50) | 0.31–0.37 (0.34) |
| Cherry Cross | 4.41–4.58 (4.50) | 0.33–0.37 (0.35) | 2.13–2.37 (2.25) | 2.00–2.70 (2.37) | 0.48–0.56 (0.53) | 0.28–0.33 (0.30) |
| Cherry Wine | 3.75–4.68 (4.27) | 0.29–0.42 (0.38) | 2.08–2.62 (2.40) | 2.94–4.95 (4.05) | 0.48–0.69 (0.60) | 0.27–0.35 (0.31) |
| Cherry Cross × Cherry Wine | 3.72–4.19 (3.91) | 0.26–0.34 (0.30) | 2.13–2.65 (2.46) | 2.79–3.61 (3.18) | 0.58–0.61 (0.59) | 0.27–0.32 (0.29) |
| Early Pearly | 4.57–4.98 (4.76) | 0.33–0.40 (0.37) | 2.62–2.89 (2.79) | 3.83–5.34 (4.76) | 0.38–0.50 (0.45) | 0.30–0.34 (0.32) |
| Electra | 4.56–4.70 (4.65) | 0.33–0.36 (0.34) | 2.09–2.23 (2.16) | 1.50–1.96 (1.77) | 0.37–0.44 (0.41) | 0.31–0.35 (0.33) |
| Endurance | 3.77–4.44 (4.13) | 0.30–0.38 (0.34) | 2.87–3.41 (3.06) | 3.68–4.93 (4.21) | 0.42–0.54 (0.47) | 0.29–0.33 (0.32) |
| Midwest | 4.76–4.93 (4.82) | 0.37–0.41 (0.40) | 2.74–3.06 (2.88) | 2.33–2.96 (2.58) | 0.47–0.55 (0.50) | 0.31–0.34 (0.32) |
| Stout | 4.37–4.64 (4.49) | 0.38–0.43 (0.40) | 2.45–2.64 (2.55) | 1.63–2.05 (1.80) | 0.42–0.51 (0.46) | 0.30–0.33 (0.32) |
| Suver Haze | 3.29–4.36 (3.75) | 0.27–0.39 (0.33) | 2.65–2.76 (2.72) | 2.23–3.56 (2.80) | 0.40–0.58 (0.50) | 0.25–0.32 (0.28) |
| Sweetened | 3.84–4.09 (3.96) | 0.36–0.39 (0.38) | 2.20–2.33 (2.28) | 1.51–1.68 (1.62) | 0.34–0.38 (0.36) | 0.27–0.29 (0.28) |
| [1] *Significance* | *** | ** | *** | *** | *** | ** |
| [2] *MSD* | 0.58 | 0.08 | 0.39 | 1.29 | 0.14 | 0.05 |
| [3] **Survey Range** | **3.29–4.98 (4.30)** | **0.26–0.43 (0.36)** | **2.08–3.41 (2.57)** | **1.50–5.34 (2.91)** | **0.34–0.69 (0.48)** | **0.25–0.37 (0.31)** |

[1] ** or *** indicate statistically significant differences among sample means based on F test at $p \leq 0.05$, $p \leq 0.01$, or $p \leq 0.001$ respectively. [2] Minimum significant difference according to Tukey's (HSD) Test at $p < 0.05$. [3] Range of survey values across all thirteen cultivars followed by means in parentheses.

**Table 2.** Comparison of combined macronutrient concentration ranges and means for the thirteen greenhouse-propagated cannabidiol (CBD) hemp cultivars to reference survey ranges with additional reference to deficiency thresholds. Combined survey range values to indicate the minimum and maximum expanded nutrient acquisition ranges among all referenced studies.

| Reference Range | Macronutrient (%) | | | | | |
|---|---|---|---|---|---|---|
| | N | P | K | Ca | Mg | S |
| [1] Survey Range | 3.29–4.98 (4.30) | 0.26–0.43 (0.36) | 2.08–3.41 (2.57) | 1.50–5.34 (2.91) | 0.34–0.69 (0.48) | 0.25–0.37 (0.31) |
| [2] Reference Survey Range | 2.65–4.47 (3.75) | 0.31–0.44 (0.35) | 1.54–2.98 (2.42) | 0.53–2.14 (1.15) | 0.25–0.46 (0.32) | 0.19–0.29 (0.24) |
| [3] Reference Survey Range | 3.30–4.76 | 0.24–0.49 | 1.83–2.35 | 1.47–4.42 | 0.40–0.81 | 0.17–0.26 |
| [4] Deficiency Threshold | 1.62 | 0.09 | 0.41 | 0.39 | 0.12 | 0.11 |
| [5] **Nutrient Acquisition Ranges** | **2.65–4.98** | **0.24–0.49** | **1.54–3.41** | **0.53–5.34** | **0.25–0.81** | **0.17–0.41 \*** |

[1] Survey values across all thirteen cultivars in this study with means in parentheses. [2] Survey ranges with means in parentheses as reported by Landis et al. [11] [3] Survey ranges as reported by Bryson et al. [10] [4] Deficiency thresholds as reported by Cocksen et al. [12] [5] Combined ranges from all surveys to include minimum and maximum nutrient acquisition levels. * indicates a higher maximum concentration value reported in control plants of a nutrient disorder study, which exceeded all survey range values [12].

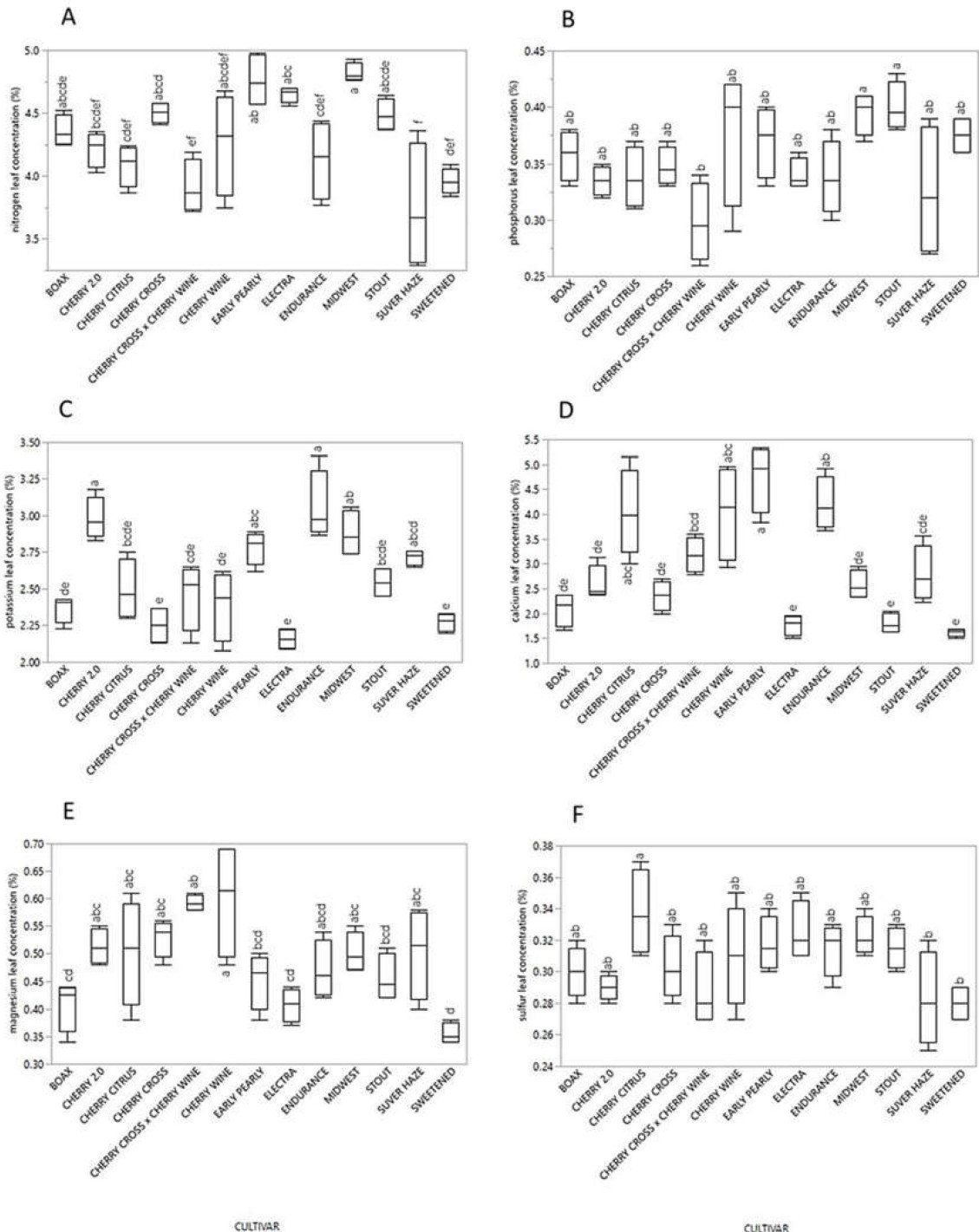

**Figure 1.** Leaf tissue macronutrient concentration ranges for thirteen greenhouse-propagated cannabidiol (CBD) hemp cultivars. Plants were 12 weeks-old and in a vegetative stage of development prior to the excision of initial cuttings for propagation. (**A**) nitrogen, (**B**) phosphorus, (**C**) potassium, (**D**) calcium, (**E**) magnesium, and (**F**) sulfur. Nutrient means that share similar lowercase letters within each graph are not significantly different from each other according to Tukey's (HSD) Test at *p* > 0.05.

*3.2. Micronutrients*

Significant differences were identified for mean micronutrient concentrations among all 13 cultivars (Table 3). Iron (Fe) minimum concentration was found to be within the range reported by Landis et al. [11], but two cultivars ("Cherry Wine" and "Midwest") exceeded the maximum level

of 150 mg·kg$^{-1}$ reported by Bryson et al. [10], with concentrations at 169 mg·kg$^{-1}$ and 152 mg·kg$^{-1}$, respectively. Twelve of the thirteen cultivars exceeded the manganese (Mn) maximum concentration of 93 mg·kg$^{-1}$ reported by Bryson et al. [10] (Table 4), with a mean of 140 mg·kg$^{-1}$ across all 13 cultivars. Two cultivars ("Early Pearly" and "Stout") exceeded the zinc (Zn) maximum reported value of 52 mg·kg$^{-1}$ by Bryson et al. [10], with 53.9 mg·kg$^{-1}$ and 54.9 mg·kg$^{-1}$, respectively. One cultivar ("Cherry Cross") was found to have a lower copper (Cu) concentration (1.6 mg·kg$^{-1}$) than the reported minimum (1.8 mg·kg$^{-1}$) and a higher mean of 4.6 mg·kg$^{-1}$ across all cultivars compared to 3.5 mg·kg$^{-1}$ reported by Landis et al. [11] Boron (B) concentrations across all 13 cultivars were found to be within the reported minimum value reported by Landis et al. [11] and within the maximum reported value of Bryson et al. [10] "BaOx" and "Cherry Cross" were found to have lower minimum aluminum (Al) concentration values of 0.1 mg·kg$^{-1}$ and 0.6 mg·kg$^{-1}$ than the minimum concentration of 0.68 mg·kg$^{-1}$ reported by Landis et al. [11] The mean across all cultivars was found to be higher at 11.1 mg·kg$^{-1}$ than reported by Landis et al. [11] with a mean of 5.58 mg·kg$^{-1}$.

**Table 3.** Leaf tissue micro-nutrient concentration ranges for thirteen greenhouse-propagated cannabidiol (CBD) hemp cultivars. Means are delineated in parenthesis following each cultivar nutrient range. Plants were 12 weeks old and in a vegetative stage of development prior to the excision of initial cuttings for propagation.

| Cultivar | Micronutrient (mg·kg$^{-1}$) | | | | | |
|---|---|---|---|---|---|---|
| | Fe | Mn | Zn | Cu | B | Al |
| BaOx | 96.5–105.0 (101.6) | 91.4–125.0 (114.6) | 31.3–39.2 (35.2) | 5.6–6.3 (5.9) | 41.3–49.2 (46.3) | 0.1–4.8 (1.9) |
| Cherry 2.0 | 114.0–131.0 (125.0) | 156.0–181.0 (170.3) | 40.3–45.7 (42.9) | 4.1–4.6 (4.4) | 41.2–47.9 (45.8) | 17.9–28.2 (21.4) |
| Cherry Citrus | 90.6–124.0 (111.2) | 120.0–203.0 (156.5) | 35.5–37.7 (36.4) | 4.7–5.2 (4.9) | 38.0–51.0 (43.7) | 4.3–9.6 (6.0) |
| Cherry Cross | 83.5–118.0 (104.4) | 83.6–143.0 (115.2) | 26.4–38.0 (31.8) | 1.6–4.9 (3.7) | 37.4–46.6 (42.9) | 0.6–8.0 (5.6) |
| Cherry Wine | 95.1–169.0 (119.5) | 122.0–183.0 (157.0) | 34.2–36.6 (35.1) | 2.8–4.8 (3.6) | 31.4–38.7 (36.1) | 5.0–20.2 (11.3) |
| Cherry Cross × Cherry Wine | 116.0–133.0 (122.3) | 135.0–174.0 (150.5) | 38.1–41.6 (40.0) | 3.5–4.1 (3.7) | 37.3–43.7 (41.3) | 7.8–26.4 (18.2) |
| Early Pearly | 113.0–135.0 (128.0) | 233.0–264.0 (250.0) | 37.5–53.9 (46.4) | 5.4–7.0 (6.1) | 68.6–90.5 (82.2) | 2.4–10.3 (5.5) |
| Electra | 97.2–115.0 (107.6) | 89.9–108.0 (98.0) | 37.8–43.9 (39.8) | 3.9–4.4 (4.3) | 25.8–29.3 (27.6) | 1.1–5.3 (2.7) |
| Endurance | 104.0–122.0 (112.3) | 144.0–189.0 (167.3) | 42.6–46.8 (44.9) | 3.6–4.5 (4.1) | 52.9–61.7 (57.7) | 8.9–15.6 (11.2) |
| Midwest | 130.0–152.0 (139.3) | 118.0–170.0 (143.0) | 47.0–52.0 (49.6) | 4.0–4.5 (4.3) | 41.1–50.0 (44.8) | 11.6–25.9 (20.0) |
| Stout | 92.6–110.0 (101.4) | 86.4–100.0 (93.9) | 38.3–54.9 (45.1) | 4.3–5.4 (4.9) | 30.4–33.0 (31.5) | 7.7–21.3 (13.5) |
| Suver Haze | 106.0–125.0 (114.5) | 118.0–165.0 (134.5) | 37.8–46.7 (43.2) | 3.8–6.0 (5.0) | 34.0–48.3 (39.9) | 4.6–33.6 (17.3) |
| Sweetened | 96.1–99.7 (97.4) | 66.5–71.3 (69.3) | 38.4–51.4 (44.0) | 3.7–4.9 (4.3) | 29.4–31.2 (30.0) | 6.3–13.3 (10.0) |
| [1] *Significance* | ** | *** | *** | *** | *** | *** |
| [2] *MSD* | *32.0* | *48.4* | *9.9* | *1.6* | *11.3* | *14.3* |
| [3] **Survey Range** | **83.5–169.0** (114.2) | **66.5–264.0** (140.0) | **26.4–54.9** (41.1) | **1.6–7.0** (4.6) | **25.8–90.5** (43.8) | **0.1–33.6** (11.1) |

[1] ** or *** indicate statistically significant differences among sample means based on F test at $p \leq 0.05$, $p \leq 0.01$, or $p \leq 0.001$ respectively. [2] Minimum significant difference according to Tukey's (HSD) Test at $p < 0.05$. [3] Survey alues across all thirteen cultivars followed by means in parentheses.

**Table 4.** Comparison of combined micronutrient concentration ranges and means for the thirteen greenhouse-propagated cannabidiol (CBD) hemp cultivars to reference survey ranges with additional reference to deficiency and toxicity thresholds. Combined survey range values to indicate the minimum and maximum expanded nutrient acquisition ranges among all referenced studies.

| Reference Range | Micronutrient (mg·kg⁻¹) | | | | | |
|---|---|---|---|---|---|---|
| | **Fe** | **Mn** | **Zn** | **Cu** | **B** | **Al** |
| [1] Survey Range | 83.5–169.0 (114.2) | 66.5–264.0 (140.0) | 26.4–54.9 (41.1) | 1.6–7.0 (4.6) | 25.8–90.5 (43.8) | 0.1–33.6 (11.1) |
| [2] Reference Survey Range | 59.0–132.0 (82.2) | 24.3–71.9 (37.1) | 23.2–46.2 (31.0) | 1.8–11.4 (3.5) | 22.6–57.3 (35.9) | 0.68–71.0 (5.58) |
| [3] Reference Survey Range | 100.0–150.0 | 41.0–93.0 | 24.0–52.0 | 5.0–7.1 | 56.0–105.0 | [†] NR |
| [4] Deficiency Threshold | 60.1 | 7.56 | 10.7 | 1.41 | 2.46 | NR |
| [4] Toxicity Threshold | NR | 47.9 | NR | NR | 671.8 | NR |
| [5] **Nutrient Acquisition Ranges** | **59.0–169.0** | **24.3–264.0** | **23.2–54.9** | **1.6–11.4** | **22.6–105.0** | **0.1–71.0** |

[1] Survey values across all 13 cultivars with means in parentheses. [2] Survey ranges with means in parentheses as reported by Landis et al. [11] [3] Survey ranges as reported by Bryson et al. [10] [4] Deficiency and toxicity thresholds as reported by Cocksen et al. [12] [5] Combined ranges from all surveys to include minimum and maximum nutrient acquisition levels. [†] Value is not reported.

Micronutrient mean comparisons using Tukey's HSD test of the 13 cultivars showed significant differences between cultivar means (Figure 2). "Midwest" had a higher Fe mean compared to "BaOx", "Cherry Cross", "Stout", and "Sweetened" with an MSD ≥ 32 mg·kg⁻¹. The highest Mn mean was in "Early Pearly", elevated from all other cultivars with an MSD ≥ 48.4 mg·kg⁻¹. "Early Pearly" and "Midwest" were higher in Zn mean concentrations compared with "BaOx", "Cherry Citrus", "Cherry Cross", and "Cherry Wine", with an MSD ≥ 9.9 mg·kg⁻¹. "Cherry Cross", "Cherry Wine", "Cherry Cross × Cherry Wine", and "Endurance" contained lower than average Cu mean concentrations across all cultivars and were dramatically lower than accumulation in "Early Pearly" with an MSD ≥ 1.6 mg·kg⁻¹. Boron mean concentration was highest in "Early Pearly" and was found to be almost 2× higher than all cultivars with an MSD ≥ 11.3 mg·kg⁻¹. Aluminum mean concentration was highest in "Cherry 2.0" and notably higher than "BaOx", "Cherry Citrus", "Cherry Cross", "Early Pearly", and "Electra" with an MSD ≥ 14.3 mg·kg⁻¹.

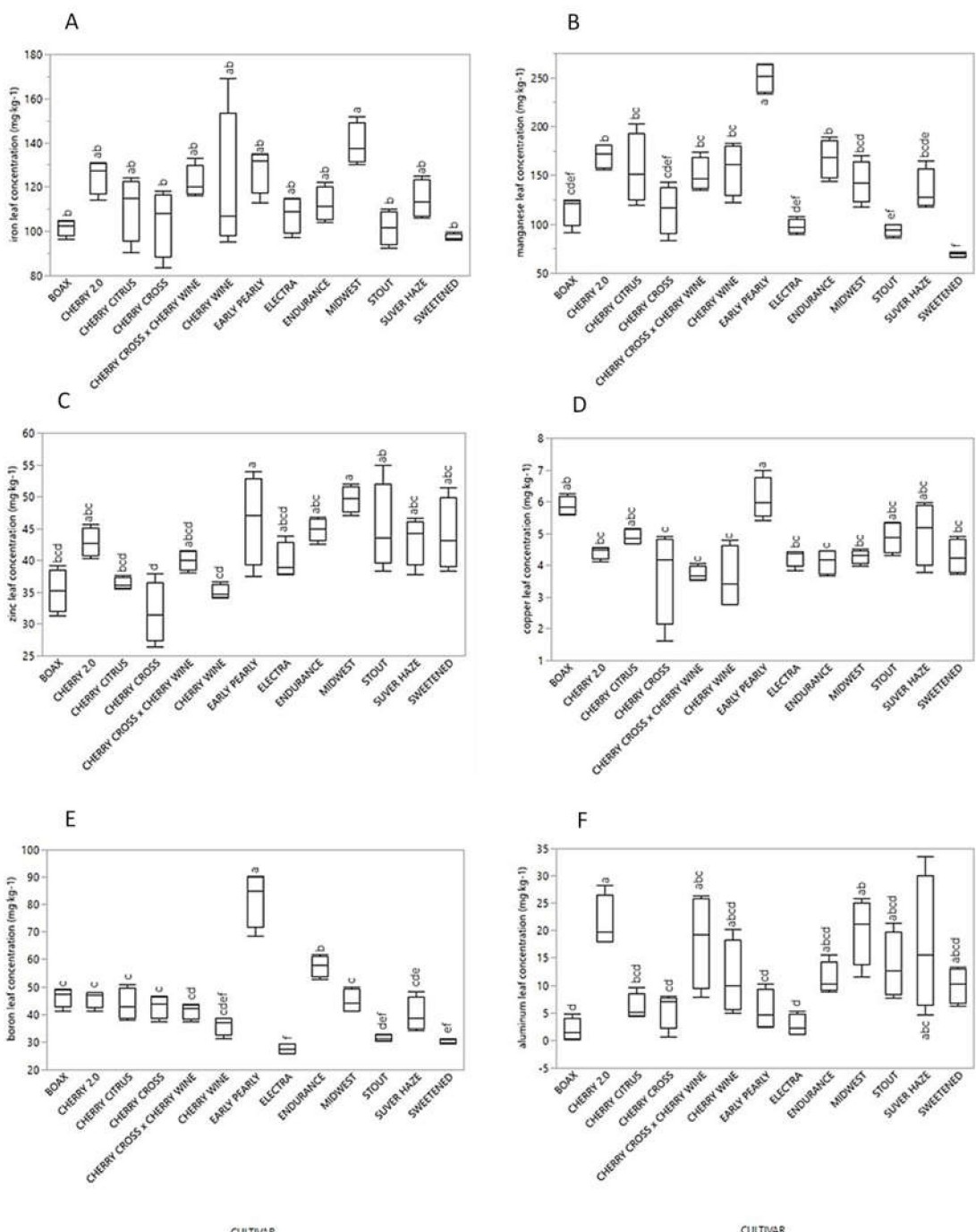

**Figure 2.** Leaf tissue micronutrient concentration ranges for thirteen greenhouse-propagated cannabidiol (CBD) hemp cultivars. Plants were 12 weeks old and in a vegetative stage of development prior to the excision of initial cuttings for propagation. (**A**) Iron, (**B**) manganese, (**C**) zinc, (**D**) copper, (**E**) boron, and (**F**) aluminum. Nutrient means that share similar lowercase letters within each graph are not significantly different from each other according to Tukey's (HSD) Test at $p > 0.05$.

## 4. Discussion

The current analysis of nutrient acquisition ranges expands the previously reported ranges of greenhouse-grown hemp cultivars by Landis et al. [11] and nursery production plants by Bryson et al. [10] (Tables 2 and 4) Our findings suggest that the maximum macronutrient concentration should be increased for N (4.98%), K (3.41%), Ca (5.34%), and S (0.37%). Additionally, micronutrient leaf tissue analyses

indicated that maximum concentrations should be increased for Fe (169 mg·kg$^{-1}$), Mn (264 mg·kg$^{-1}$), and Zn (54.9 mg·kg$^{-1}$), while ranges for Cu and Al should include minimum values of 1.6 mg·kg$^{-1}$ and 0.1 mg·kg$^{-1}$. Leaf tissue concentrations of P, Mg, and B were within previously reported ranges of both referenced survey values by Bryson et al. [10] and Landis et al. [11].

Baseline nutritional information is limited for many specialty crops. Initial data by Bryson et al. [10] provided a starting point, but their report was not specific about how many plants were sampled, nor how many leaves were sampled from individual plants, only indicating that 25 mature leaves were taken from new growth in the vegetative stage. It is also unknown how many cultivars were represented in the Bryson et al. [10] survey or if different subspecies [2,14] were included. Nutrient ranges reported in the more recent survey by Landis et al. [11] were analyzed using five *Cannabis sativa* hemp cultivars with five replicates per cultivar for a total of 25 samples. Samples in the Landis et al. [11] survey were taken at the 12-week vegetative stage prior to flowering comparable to the current survey which expands both the sample size to 52 and the number of cultivars sampled to 13.

Three of the cultivars utilized in our study (Endurance, Stout, and Sweetened) were also represented in the Landis et al. [11] nutrient survey. These cultivars exhibited some elevated nutrient acquisition ranges compared to the previous survey work by Landis et al. [11], particularly for elements such as Ca, Mn, Zn, and Boron. Calcium concentrations were most notably higher within the cultivar Endurance at 4.21% compared to the previous reported value of 0.73% [11], but lower Ca concentrations were observed in the cultivar "Sweetened" at 1.62% compared to the previously reported 2.03% [11]. Elemental nutrients Mn and Zn were higher for all three cultivars in the present study and B concentration varied between studies [11].

One explanation for these differences could be related to biomass, since the plants in this study were smaller and were grown in the spring season, while in the previous study, conducted by Landis et al. [11], the plants were larger and were grown later in the season. This implies that there may be a dilution effect based on dry weight as larger plants will spread out the nutrients in more tissue over time as discussed by Bryson et al. [10] However, the samples in the current study were only utilized to determine a level of nutrient accumulation in leaf tissue by a targeted survey range and samples were taken from actively growing stock plants. As such, we did not compare effects that would benefit from dry weight measurements and acknowledged that further research should be conducted when evaluating catalysts that may contribute to differences found between survey ranges of replicate cultivars.

Recent work conducted by Cockson et al. [12], in characterization of nutrient disorders of *Cannabis sativa*, lends an additional set of ranges of analysis of leaf tissue concentrations from the study's control plants (data not shown). Ranges in the work conducted by Cockson et al. [12] predominantly fell within the scope of our survey range analysis, and though the primary objective was not the expansion of these values, there were some additive data applicable to expanding previously reported concentration ranges. One nutrient of interest reported by Cockson et al. is S, with a maximum accumulation found to be 0.41% in control plants, higher than the reported published data, and our finding of 0.37% (Table 2). Additionally, Fe deficiency symptoms reported by Cockson et al. [12] were visually evidenced at a leaf nutrient concentration of 60.1 mg·kg$^{-1}$. However, ranges reported by Landis et al. [11] designate a minimum nutrient concentration for Fe of 59 mg·kg$^{-1}$ without visual symptoms of nutrient deficiency, slightly below the level reported for Fe deficiency. Manganese toxicity was noted to occur at a concentration of 47.9 mg·kg$^{-1}$ [12], but this level is within all previously reported survey ranges [11,12] and below the minimum value identified in the current survey analysis. The modest difference in the minimum threshold for Fe and toxicity level for Mn could perhaps be exclusive to cultivar type or subspecies [2,14] and should be further evaluated. It should also be noted that only one cultivar ("T1") was utilized in the nutrient disorder study conducted by Cockson et al. [12], and although "T1" was also included in the five cultivars utilized by Landis et al. [11], this cultivar was not represented in the sampling pool of the current survey range analysis.

The fertility survey ranges evaluated in this study illuminated significant differences among cultivars in nutrient acquisition, thus suggesting that there may be physiological differences among *Cannabis sativa* hemp cultivars propagated in a controlled environment that could potentially affect nutrient uptake and secondary metabolite production. Since cultivars utilized in this study included both subspecies [2,14] *C. sativa* and *C. indica*, a brief analysis was conducted to determine any significant differences, with cultivars "Midwest", "Cherry 2.0", and "Early Pearly" belonging to the latter. Anatomical features, specifically height, was also considered in these comparisons between subspecies [2,14]. Focus was placed on N, P, and K for general nutrient acquisition comparisons.

"Midwest" and "Early Pearly" had the highest mean accumulation value for leaf tissue N concentrations (4.82% and 4.76%) than taller cultivars belonging to subspecies [2,14] *C. sativa* ("Cherry Citrus", 4.09%; "Cherry Wine", 4.27%; "Electra", 4.65%; and "Endurance", 4.13%). "Midwest" had less within-sample variation for N concentration between other *C. indica* subspecies [2,14] but was found to be comparable to within sample variations of *C. sativa* cultivars such as the tall "Electra" cultivar and the shorter "Cherry Cross" cultivar. "Midwest" also had the highest mean accumulation of P at 0.40% within the subspecies [2,14] group *C. indica*, but this was comparable to the mean P accumulation of "Stout", a short *C. sativa* type. "Cherry 2.0" had less within-sample variation of P among *C. indicas*, but this was found to be comparable to the tall "Electra" and the short "Sweetened" *C. sativa* cultivars. Comparisons were made to the tall *C. sativa* cultivar, "Cherry Wine", noted to also have a high mean accumulation of P (0.38%), but showed high variation within the sample. Analysis of K concentrations demonstrated *C. indica* types to have a higher mean accumulation in leaf tissue ("Cherry 2.0", 2.98%; "Early Pearly'" 2.79%; and "Midwest", 2.88%) than *C. sativa* types, except for "Endurance" (3.06%), a tall *C. sativa* cultivar. However, there was high variation within the sample of "Endurance" at 0.54%. Though no significant differences could be noted in this study to substantiate dissimilarities between *C. sativa* and *C. indica* types, or anatomical differences of height in nutrient acquisition, further research should be conducted to conclusively rule out potential dissimilarities, while establishing additive nutrient survey ranges for various cultivars of *Cannabis*.

## 5. Conclusions

The analysis of leaf tissue concentrations for both macro- and micronutrients in this survey found significant differences among CBD cultivars being used as vegetative mother stock prior to the harvesting of cuttings in a greenhouse setting. The acquired survey ranges exceed those previously reported, broadening the scope of fertility ranges for *Cannabis sativa* hemp cultivars. The survey ranges observed in this study suggest there are differences in acquisition and partitioning of nutrients based on the cultivar, and potentially subspecies, of the *Cannabis sativa* plant. Further research should be conducted to evaluate these dissimilarities in their entirety in addition to deficiency and toxicity thresholds to promote enhanced fertility management strategies of greenhouse cultivated hemp clonal varieties targeted for CBD production.

**Supplementary Materials:** The following are available online at http://www.mdpi.com/2311-7524/6/4/98/s1, Spreadsheet S1: Nutrient Survey Ranges Raw Data.

**Author Contributions:** Conceptualization: B.E.W. and M.M.; Methodology: J.K., B.E.W., P.V. and P.C.; Software: B.E.W., P.C. and J.K.; Validation: J.K., K.E., J.D. and M.M.; Formal analysis: J.K., P.C., B.E.W. and M.M.; Investigation: J.K., P.V. and B.E.W.; Resources: M.M., J.K., P.C. and B.E.W.; Data curation: J.K. and B.E.W.; Writing—original draft preparation: J.K.; Writing—review and editing: K.E., J.D., M.M., K.H. and B.E.W.; Visualization: J.K.; Supervision: K.E., J.D., B.E.W., M.M. and K.H.; Project administration, J.K. and B.E.W.; Funding acquisition: K.E. and B.E.W. All authors have read and agreed to the published version of the manuscript.

**Funding:** This research received no external funding.

**Acknowledgments:** We would like to thank the North Carolina Department of Agriculture and Consumer Services for assistance with tissue analysis testing with this research.

**Conflicts of Interest:** The authors declare no conflict of interest.

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
