# Peer review of "Augmenting Nutrient Acquisition Ranges of Greenhouse Grown CBD (Cannabidiol) Hemp (Cannabis sativa) Cultivars"

_horticulturae, doi:10.3390/horticulturae6040098_

Round 1
Reviewer 1 Report
This works does help to fill a void of information on hemp cultivars. Suggest a few wording changes to clarify the results.

Reviewer 2 Report
I believe that the article is correctly written. Provides essential information on getting to know the cannabis plants anew. The information contained in the article may be useful for other scientists in the search for new varieties, carrying out such cultivation technology and obtaining the necessary compounds. The article may be published as is.
Author Response
Thank you for the review.
Reviewer 3 Report
I recommend increasing only the background and references in the introduction.Author Response
Thank you for the review.
Spell check has been completed throughout the document.
We understand the comment(s) about the concise literature review. This often occurs when limited scientific studies have been conducted in a specialized area of research or in the case of Cannabis, one that was until recently illegal. We choose to take a more direct and concise approach to the literature review. While there are numerous other citations that could be added to increase the length of the literature review, it would only be tangential additions. Hence, we decided to limit our discussion to directly applicable studies. Therefore, we decided not to increase the length of the literature review.
Reviewer 4 Report
Dear Author, the manuscripts appears as far as interesting and fitting with the topics of the journal. The experiment is accurate and well described, and the results are interesting and useful for the optimization of the Cannabis sativa cultivation.
I have just some suggestions:
- Keyword: avoid using the same words that are in the title
- Line 70: say why you chose these cultivars.
- Line 77: describe what you mean for “most recently mature leaves”.
- Line 86-88: add more details about the method of extraction and analyses.
- Tables: use only the mean value ± the standard error.
- Line 95-112: move this part to the Discussion.
- Line 149-163: move this part to the Discussion.
- Table 1 and Figure 1: If I am not wrong, there are reported the same data, please use only the tables or the figures to show these data.
- Table 3 and Figure 2: If I am not wrong, there are reported the same data, please use only the tables or the figures to show these data.
- Line 202-205: use the same unit (% or mg kg-1) for all the values.
- Line 208-235: shorten this part moving some sentence in the introduction, leave only the sentences need to discuss the results obtained in the present work.
- Line 268: remove citations from the Conclusions
Kind regards
Author Response
Thank you for your review. Please see our reply attached.
